# ROMUL: Scale Adaptative Population Based Training

## Abstract

In most pragmatic settings, data augmentation and regularization are essential, and require hyperparameter search. Population based training (PBT) is an effective tool for efficiently finding them as well as schedules over hyperparameters. In this paper, we compare existing PBT algorithms and contribute a new one: ROMUL, for RObust MULtistep search, which adapts its stepsize over the course of training. We report competitive results with standard models on CIFAR (image classification) as well as Penn Tree Bank (language modeling), which both depend on heavy regularization. We also open-source `hoptim`, a PBT library agnostic to the training framework, which is simple to use, reentrant, and provides good defaults with ROMUL.

## 1 Introduction

Hyperparameter tuning is essential for good performance in most machine learning tasks, and poses numerous challenges. First, optimal hyperparameter values can change over the course of training (schedules), e.g. for learning rate, fine tuning phases, data augmentation. Hyperparameters values are also rarely independent from each other (e.g. the magnitude of individual data augmentations depends on the number of data augmentations applied), and the search space grows exponentially with the number of hyperparameters. All of that search has to be performed within a computational budget, and sometimes even within a wall-clock time budget (e.g. models that are frequently re-trained on new data), requiring efficient parallelization. In practice, competitive existing methods range from random search (Bergstra & Bengio, 2012) to more advanced methods (that aim at being more compute-efficient) like sequential search (Bergstra et al., 2011; 2013; Li et al., 2018), population based training (PBT, e.g. Jaderberg et al. (2017); Ho et al. (2019)) and search structured by the space of the hyperparameters (Liu et al., 2018; Cubuk et al., 2019b).

A major drawback of advanced hyperparameter optimization methods is that they themselves require attention from the user to reliably outperform random search. In this work, we empirically study the different training dynamics of data augmentation and regularization hyperparameters across vision and language modeling tasks, in particular for multistep (sequential) hyperparameter search. A common failure mode (i) is due to hyperparameters that have a different effect on the validation loss in the short and long terms, for instance using a smaller dropout often leads to faster but worse convergence. Another common problem (ii) is that successful searches are constrained on adequate "hyper-hyperparameters" (such as value ranges or the search policy used, which in current methods are non-adaptive mutation steps). Our contributions can be summarized as follows:

- We present a robust algorithm for leveraging population based training for hyperparameter search: ROMUL (RObust MULtistep) search, which addresses (i) and (ii). We empirically study its benefits and limitations, and show that it provides good defaults that compare favorably to existing methods.

- We open-source `hoptim`, a simple library for sequential hyperparameter search, that provides multiple optimizers (including ROMUL), as well as toy benchmarks showcasing hyperparameter optimization problems we identified empirically and standard datasets.

## 2 HYPERPARAMETER OPTIMIZATION WITH POPULATION-BASED TRAINING

In this article, we refer to the family of algorithms that continuously tunes hyperparameters of a set of models over the course of their training as "PBT algorithms" or "PBT optimizers". Hyperparameter optimization is thus a zero order optimization performed at a slower frequency than the (often first order, e.g. SGD) optimization of the model. A *PBT step* happens typically after a fixed number of epochs or updates of the model, often optimizing the loss from the validation set, continuing from an already produced "parent" checkpoint, and producing and evaluating a new checkpoint. At every PBT step, hyperparameters can be updated (*mutated*), incremented or decremented by some number (*step size*), or sampled.

There are multiple aspects to consider when designing a PBT algorithm. *Technical constraints*: how the optimization is distributed, with a centralized or decentralized algorithm, workers to run the trainings, how failed workers are handled. They are solved in a unified manner in the experiments we performed, by the `hoptim` library to implement and compare multiple algorithms. It is decoupled from the scheduling of the jobs and designed to accommodate adding more workers to scale up the training, or fewer when some are killed, for example through preemption or time-out on a shared cluster. *Optimization method*: how the hyper-parameters are modified throughout the training, for instance through mutations. *Selection process*: which individual of the population are kept, both in term of hyper-parameters and state of the neural network (checkpoint). For those last two points, some solutions are described below.

### 2.1 CHALLENGES

In order to have a clearer understanding of our proposed methods, we show below the main concerns we have observed in PBT:

**Anisotropy**: by definition, the optimal value of the hyperparameters considered is unknown, and oftentimes the range (or mutation scheme) provided to the algorithm is a loose estimate only. As modifying two hyperparameters with the same step size can produce effects with very different magnitudes, the user is required to to normalize the search space. But pre-tuning the hyperparameter tuner itself can be cumbersome as dynamics evolve during training. Section 3.1 provides an example based on the Rosenbrock function which illustrates this issue and highlights the interest of adaptive mutations.

**Checkpoint *vs.* hyperparameters**: comparing individuals in the population is extremely hard as improvements can be due to better hyperparameters, or better checkpoints (including potentially better batches). Better performance through better checkpoints is an optimization phenomenon (e.g. random restarts), that can bias the hyperparameter selection. We will detail this aspect in Section 4.2.

**Short-term-long-term discordance**: we observed empirically that hyperparameters which induce better performance in the short term are not always optimal in the longer term. This is a challenge that does not exist in classical static optimization, but is crucial for PBT since local minima are easy to reach and pose a danger for greedy algorithms. An example of such a parameter is the learning rate. Dropping the learning rate often induces a drop in the validation loss, even early in the training, and increasing it has the opposite effect, causing greedy PBT algorithms to reduce it to the minimum value too early, without being able to recover. We will detail this aspect in Section 4.1.

### 2.2 DIFFERENTIAL EVOLUTION AND ROMUL

Differential Evolution Storn & Price (1997) (DE) is a standard black-box optimization method, for minimizing $f : \mathbb{R}^n \to \mathbb{R}$. It operates on a population $x^i \in \mathbb{R}^n$ for all $i \in \{1, ..., M\}$, $M \geq 4$, and indefinitely repeats the following steps for each individual $x^{\text{base}}$ in the population to generate another individual called *mutated vector* that could replace $x^{\text{base}}$ if better:

1. given the best individual $x^{\text{best}}$ which minimizes $f$ in the population, as well as two randomly selected ones $x^a$ and $x^b$, compute the donor $d$, which will give part of its coefficients to the mutated vector. In the current-to-best/1 scheme we use, these are the base coefficients plus a term attracting to the best set of coefficients from the current population, and an

additional random variation (a standard value for $F_i$ is $F_1 = F_2 = 0.8$):

$$d = x^{\text{base}} + F_1(x^{\text{best}} - x^{\text{base}}) + F_2(x^b - x^a) \tag{1}$$

2. create the new mutated vector $\widetilde{x}^{base}$ by randomly selecting each component $j \in \{1, ..., n\}$ of the base $x^{\text{base}}$ or the donor $d$ through the binary crossover operator: $\widetilde{x}_j^{\text{base}} = \text{CHOICE}(x_j^{\text{base}}, d_j)$. This non-linear operation lets the optimization leave the vector span of the population.

3. compute $f(\widetilde{x}^{\text{base}})$ and replace $x^{\text{base}}$ by $\widetilde{x}^{\text{base}}$ within the population if and only if $f(\widetilde{x}^{\text{base}}) \leq f(x^{\text{base}})$.

This method is interesting as it is already based on a population and adapts well to parameters with different dynamics while being simple and fully parallelizable. In particular, it does not rely on mutation ranges or step sizes - Equation 1 samples new parameters close to the current population, and as the population individuals go through selection this sampling is refined and becomes sharper around optimal values. In practice, if a parameter's bounds are too loose or wrong, DE will eventually adapt after iterations of selection by removing individuals too far from the optimal value, and concentrate its computation budget on relevant values for this parameter.

In order to use it for PBT, the set of hyper-parameters is converted to a vector in $\mathbb{R}^n$ using *nevergrad* parametrization system (Rapin & Teytaud, 2018). However, this basic version of differential evolution (also implemented in *nevergrad*) is not adapted to PBT. Indeed the training function $f$ changes with the checkpoint as we are updating the parameters (not the hyperparameters) with a stochastic gradient from the task loss. The trend of $f$ is therefore typically downwards during the training, younger generations/later epochs tending to have a lower loss than their parents', biasing the hyperperameter selection process in favor of those of the children (later steps of SGD updates) instead of in favor of better hyperparameters.

**ROMUL** We therefore propose an adaptation: a population of $n$ individuals is trained, after finishing their step, individuals are compared to the rest of the population. If they have one of the $n/k$ best loss (we use $k = 2$ throughout), the training continues without changing the hyperparameters, otherwise, the hyperparameters are mutated. If the hyperparameters of an individual are mutated $m$ times in a row (we use $m = 3$ throughout), its checkpoint is killed and replaced by one of the $n/k$ best individuals. The values of $k$ and $m$ are hyperparameters, although we did not vary them in any experiments: $k = 2$ allows to have, on average, one alternative (mutated) version to each of the ones we keep training without hyperparameter change, and $m = 3$ proved to be robust across our experiments, to select when to discard a checkpoint. If using lower $m$ values, one should consider increasing the number of epochs per PBT step to prevent culling checkpoints too early (see Section 4.1 and 4.2).

The mutation scheme is adapted to fit this use case. In Eq. 1, $x^{\text{base}}$ and $x^{\text{best}}$ are both replaced by a randomly selected set of hyperparameters $x^c$ and $x^d$ from the best $n/2$ individuals ("rand-to-rand/1" scheme following (Storn & Price, 1997; Das & Suganthan, 2011) notations). Replacing $x^{\text{base}}$ aims at keeping the path through checkpoints unimodal, since keeping several modes with corresponding checkpoints is unnecessary. Replacing $x^{\text{best}}$ by any other "good" (top 50%) set of hyperparameters aims at avoiding early convergence, which we observed as one of the main problems during trainings. This also avoids a strong bias by a good checkpoint (more on this in 4.2). To avoid duplication of hyperparameters, we opt for making $F_1$ and $F_2$ random vectors instead of using the binary crossover non-linearity. In order to keep the initial scaling of DE, we chose $\mathbf{F_1}[i]$ uniformly distributed between $0$ and $2F$ (we use the common value for $F$ from vanilla DE: $F = 0.8$), and $\mathbf{F_2}[i] = 2F - \mathbf{F_1}[i]$, $\forall i$. This ensures that the sum $\mathbf{F_1}[i] + \mathbf{F_2}[i] = 2F$, $\forall i$, as in vanilla DE. With $\odot$ the elementwise multiplication, this yields $d = x^c + \mathbf{F_1} \odot (x^d - x^c) + \mathbf{F_2} \odot (x^b - x^a)$.

## 2.3 OTHER ALGORITHMS

In our experiments, we compare several algorithms briefly presented below. We aim to compare how effective they can be for practical use-cases of hyperparameter tuning, where the user does not want to tune the hyperparameters of its hyperparameter tuner, and desires meaningful defaults. The only input they take are the number of parallel trainings, the range of hyperparameters, and a hint for an initial value (e.g. the same value 0 for dropout values and data augmentation magnitudes).

**Initiator PBT**: We reimplemented Initiator Based Evolution, presented in Li et al. (2019). New hyper-parameters are sampled from parent hyper-parameters by adding/removing a *mutation constant* (for instance $dropoutChild = dropoutParent \pm 0.1$). A newly created checkpoint is compared to a randomly sampled checkpoint in the population: if the latter is better, the new checkpoint is discarded and the latter is forked with its hyperparameters - this ensures that only the best performing models remain eventually, and allows it to run asynchronously. For each parameter, we specify a range, and use $(hi - lo)/30$ as a mutation constant unless specified otherwise.

**Truncation Selection**: $N$ models are trained in parallel. Regularly, the $M$ worst performing models are stopped and replaced with clones of the $M$ bests, and hyperparameters are randomly perturbated. This scheme was first introduced in Jaderberg et al. (2017). In our experiments, we use $M = N/4$. For hyperparameter perturbation, we generalize the mutation scheme introduced in Ho et al. (2019): each parameter is sampled uniformly in its range $[lo, hi]$ with 20% probability, or incremented by `random.choice([-3, -2, -1, 0, 0, 1, 2, 3]) * (hi - lo) / 10` and then clipped to stay within $[lo, hi]$.

**ASHA**: This is not a PBT algorithm, but a strong hyperparameter search algorithm that we compare to. In the Asynchronous Successive Halving Algorithm (ASHA, Li et al. (2018)), hyperparameters are sampled uniformly like in Random Search, but models are evaluated early and stopped if not in the top $1/\eta$ percentile. For a given model, the first evaluation can happen after 1, $\eta$, $\eta^2$, .. steps, making this algorithm robust to hyperparameters whose optimal value does not perform well until late in the training (Section 4.1). Unlike Initiator PBT or Truncation Selection, ASHA finds constant values for hyperparameters rather than schedules. We set the reduction factor $\eta$ to 3.

## 3 EXPERIMENTS

We ran experiments on a toy optimization problem (the Rosenbrock function), CIFAR, and Penn Tree Bank, all with the same ROMUL hyperparameters to test its robustness. Each of these experiments train in around 100 to 300 epochs, and we used 1 step per epoch, so that they all have similar time scales.

### 3.1 EXAMPLE ON A TOY OPTIMIZATION PROBLEM

Current PBT mutation schemes have fixed steps and therefore do not automatically adapt to the landscape of the optimized function. This means that they are not well-suited for anisotropic problems, which often arise in real life applications since some hyperparameters may be very important to tune finely, while other do not require the same precision. To highlight this issue, we experiment below on the Rosenbrock function: $\mathcal{R}_{a,b}(x, y) = (a - x)^2 + b(y - x^2)^2$

We will aim at minimizing $\mathcal{R}_{1,100}$ through the surrogate $\mathcal{R}_{\hat{a},\hat{b}}$, with $\hat{a}$ and $\hat{b}$ two hyperparameters handled with PBT. We initialize both parameters at 20 and bound them by -12.12 and 212.12 (using integers would be a special case since actual $a$ and $b$ values are integers). This experiment can be reproduced using the `hoptim` toolbox with the command: `hop bench rosenbrock`.

While standard PBT with random steps wastes mutations on $\hat{a}$, DE is able to adapt its step-size to large steps on $\hat{a}$ until getting close, then smaller steps on $\hat{a}$ to tune $\hat{a}$ and $\hat{b}$ more finely. This is visible in Fig. 1a with ROMUL values of $\hat{a}$ converging quickly to around 1. The mutations then become sharper, while the ones for Initiator-PBT (small steps) are still too large and oscillate around the optimal value. Arguably, the mutation step could have been even smaller, but that would have slowed down the convergence, and these steps would be painful meta-parameters to tune at scale. Initiator-PBT with larger steps and Truncation selection are not displayed in this figure because their variations are too large.

The impact on the loss $\hat{\mathcal{R}}$ is then visible in Fig. 1b: Initiator PBT can't decrease past 0.2 with large steps, and 0.048 with smaller steps, since it is trapped trying to optimize $\hat{a}$ while DE is able to reach better values. Fig. 1c and 1d show the trajectory of $(x, y)$ for Truncation selection and ROMUL, with the same number of training steps. Truncation selection is hampered by more random mutations. On the other hand, ROMUL is able to reach a much lower value after exhibiting a more chaotic behavior when it initially adapts to the scale of the problem. The trajectory for both versions of Initiator-PBT can be found in Fig. 2 of the appendix. Initiator-PBT with large steps (Fig. 2a) moves

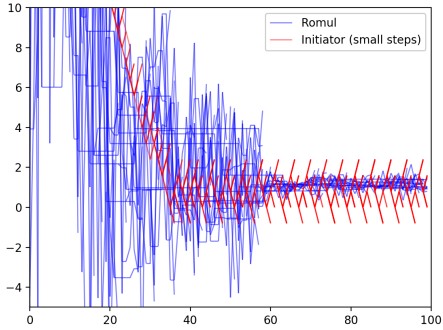 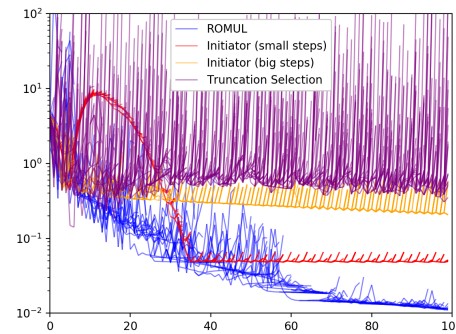

(a) Rosenbrock parameter $\hat{a}$ with respect to the number of steps (optimal at 1). After $\approx 50$ steps, ROMUL's distribution for $\hat{a}$ gets sharper around 1, Initiator always uses an hardcoded mutation step size.

(b) Loss $\mathcal{R}_{1,100}$ with respect to the number of steps (lower is better, minimum value is 0). ROMUL keeps adapting and decreasing while other optimizers are locked to higher levels depending on their step sizes.

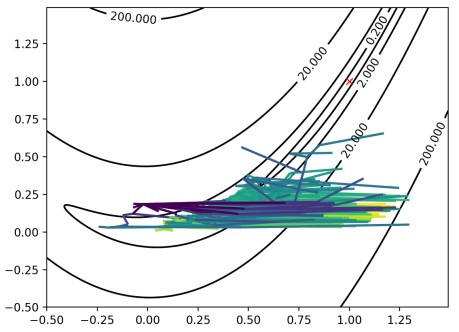 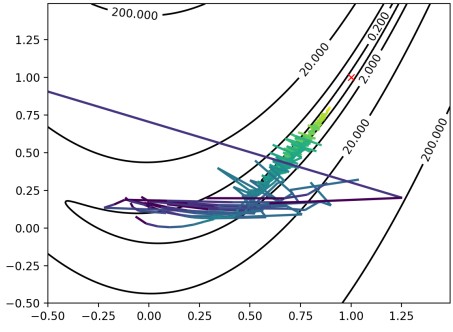

(c) Truncation Selection PBT (minimum loss=0.14)

(d) ROMUL (minimum loss=0.011)

Figure 1: Training on the Rosenbrock benchmark. ROMUL outperforms initiator and truncation selection because it can adapt its step size. Bottom plots: Trajectories of 100 PBT training steps (16 jobs per step) on the Rosenbrock function with $a = 1$ and $b = 100$ (minimum at the red cross $(1, 1)$, trajectories go from blue to green)

very slowly to the minimum because of big extra oscillations. With smaller steps (Fig. 2b), it reaches better values through a slow and non-direct path. Tab. 3 in the appendix provides quantitative results by averaging over 20 runs, including a version of Initiator PBT in which updates are performed through multiplications by 0.8 or 1.2. In particular, ROMUL performs statically better than all over optimizers on this testbed ($p < 1.1e - 5$ with a two sample Welch's t-test). In Fig. 3 in Appendix, we also show the behavior with more variables by performing optimization on an average of Rosenbrocks functions, each with independent $a$ and $b$ variable to be estimated by PBT. Overall ROMUL performs consistently well across the board for a wide range of number of variables.

## 3.2 APPLICATION TO CIFAR (IMAGE CLASSIFICATION)

In this section, we compare various algorithms for tuning hyperparameters for image classification on CIFAR (Krizhevsky et al., 2009). We reproduce the population based augmentation (PBA) setup from (Ho et al., 2019) with their original implementation. Our algorithms train a Wide-ResNet-28-10 model on Reduced CIFAR-10 (using 10%, i.e. 4000 images, of the training set for actual training, and the remainder as a validation set), and optimize the same 60 hyperparameters as in Ho et al. (2019): 2 magnitudes and 2 probabilities for each of the 15 possible data augmentations. For each algorithm, we take the best model in the validation set at epoch 200, and use its hyperparameters schedules to train another Wide-ResNet-28-10 model on CIFAR-10 and CIFAR-100 and finally report test accuracies at epoch 200 in Table 1. Trainings can be reproduced with the `hoptim`

package and its benchmarking counterpart `hoptim_benchmarks` in the `cifar` folder. ROMUL recovers most of the gains on CIFAR-10 (2.8% error vs. 2.6% for the SOTA and 3.9% for the baseline), and is a bit further away on CIFAR-100 (17.1% vs. 16.7% for the SOTA and 18.8% for the baseline). PBA, which yields state-of-the-art results on CIFAR, used Truncation selection PBT introduced in (Jaderberg et al., 2017), which we implemented and compared to. We adopted all the PBA hyperparameters and observe 2.7% on CIFAR-10 (ROMUL: 2.8%) and 17.7% on CIFAR-100 (ROMUL: 17.1%). The differences in the job and population management in `hoptim` may explain the difference between our implementation and theirs, which is particularly marked on the training set reduced CIFAR-10: 12.8% for their vs. 13.9% for our implementation.

Table 1: Classification error (lower is better) on CIFAR-10 and CIFAR-100 test sets for a Wide-ResNet-28-10 (36M params). The algorithms run with 16 workers in parallel with the same compute budget (except when stated otherwise) on reduced CIFAR-10. After that, the schedule found is used for training the same model from scratch on CIFAR-10 and CIFAR-100

| Algorithm | Reduced CIFAR-10 (10%) | CIFAR-10 | CIFAR-100 |
|---|---|---|---|
| Baseline: Wide-ResNet-28-10 | n/a | 3.9 | 18.8 |
| RandAugment (Cubuk et al., 2019b) | n/a | 2.7 | 16.7 |
| PBA (3 epochs/step) (Ho et al., 2019) | 12.8 | 2.6 | 16.7 |
| ASHA | 14.7 | 2.8 | 17.6 |
| ASHA (running for double the time) | 14.1 | 2.7 | 17.2 |
| Truncation Selection (PBA, ours) | 13.9 | 2.7 | 17.7 |
| Initiator PBT (Li et al. (2019), ours) | 14.7 | 2.9 | 17.9 |
| ROMUL | 14.0 | 2.8 | 17.1 |

As PBA waits for more than 1 epoch/step to evaluate a set of hyperparameters, we compared 1 epoch/step and 3 epochs/step (their setting), as it could help thwarting short-term/long-term discrepancy effects (see 4.1) and noise as explained above, but we could not identify a sufficiently generic scheme for all applications. In general, this is part of the PBT hyperparameters that are tuned in PBA, that we try to completely remove as hyperparameters in ROMUL, by being insensitive to it (in this case it does not seem to affect Truncation Selection either). For this hyperparameter, the constraint is to do PBT steps slow enough so that the number of updates is sufficiently large for the model to adapt to new mutated hyperparameters, and high enough so that PBT has enough steps to optimize the hyperparameters. In practice, trainings are long enough (regarding the number of SGD updates of the model) for a wide range of PBT steps frequencies to work.

### 3.3 APPLICATION TO THE PENN TREEBANK DATASET (LANGUAGE MODELING)

We experiment with the TransformerXL model (Dai et al., 2019) on the PTB dataset (Marcus, 1993). TranformerXL's code is open-source and is the state-of-the-art for tranformer models on this dataset when using proper regularization, making it an interesting challenge for PBT. It comes with several dropout hyperparameters: we search for optimal values for five different dropout hyperparameters, that we describe in Table 4 in appendix. They are all initialized to 0 with standard deviation of 0.1 for ROMUL (negative values are reflected to positive values), hence not at the baseline values.

Results are reported in Table 2. Trainings can be reproduced (up to random variance) with the `hoptim` package and its benchmarking counterpart `hoptim_benchmarks` in the `ptb` folder. The baseline TransformerXL was obtained with the author's code and is close to the one reported in the initial paper. Noticeably, ASHA and Random Search (with a uniform prior) are not able to come close to the baseline, with more than 4 points difference in both validation and test perplexity (PPL). Truncation selection and Initiator PBT on the other hand are able to reach the baseline although they were not able to excel it in test perplexity. Only ROMUL is able to reach (marginally) better results than the reproduced baseline in test PPL with both 16 and 32 workers. A found dropout schedule is displayed in the appendix (Fig. 4) and show dropouts rapidly increasing in the beginning and stabilizing to different levels. Using 16 and 32 workers provided similar results up to noise for ROMUL (in this very case, 32 workers does not actually perform better than with 16 workers).

However, using 8 workers results in a notable drop in performance for all optimizers (Test PPL ROMUL 56.39, TruncSel 57.98, Initiator 56.33).

Table 2: Perplexity (lower is better) on PTB for a Transformer-XL with 16 layers and 24M parameters, best validation PPL before iteration 175 and corresponding test PPL, given the resources needed these values are not averaged, numbers excelling our training baseline are in **bold**.

| Training | workers | Validation PPL | Test PPL |
|---|---|---|---|
| TransformerXL SOTA (Dai et al. (2019)) | 1 | / | 54.52 |
| TransformerXL SOTA (our training, their code) | 1 | 59.65 | 55.43 |
| ASHA | 16 | 63.20 | 58.35 |
| Truncation Selection PBT | 16 | 60.24 | 57.29 |
| Initiator PBT | 16 | **59.42** | 55.80 |
| ROMUL PBT | 16 | **57.83** | **55.16** |
| Random Search | 32 | 63.84 | 60.90 |
| ASHA | 32 | 64.31 | 61.63 |
| Truncation Selection PBT | 32 | **58.45** | 55.93 |
| Initiator PBT | 32 | **59.36** | 55.73 |
| ROMUL PBT | 32 | **58.63** | **55.28** |

## 4 DISCUSSION

### 4.1 SHORT TERM - LONG TERM DISCORDANCE

We have observed on PTB and other applications that some hyperparameters were never contributing positively to the model's performance in the short term (eg: 1 step) but could become better on a longer term (eg: 5 steps or more), hence checkpoints need time to adapt to a new parameter set (e.g.: building more redundancy). In an experiment on PTB, we used one epoch per step for half the training and then modified it to 10 epochs per step. We observed that increasing one of the dropout contributed negatively for small steps (1 epoch), but positively for long steps (10th epochs). This is a major roadblock for PBT-based approaches since two models with different hyperparameters can't be straightforwardly compared at every step, but only after an unknown delay. This is partially handled by being conservative on models to keep: keeping the best 50% unchanged in ROMUL, or the random tournament scheme that allows bad models to continue in Initiator PBT when assigned an even worse opponent. Fig. 5 in appendix shows such an example in another domain: a large dropout seems very detrimental early on, but very beneficial in the longer term. While this is expected for regularizations, we observe that a straightforward schedule increasing the dropout in steps (in red) is not able to compensate this - we observed the same effect with a continuous schedule. This behavior adds complexity to the task of PBT algorithms, because bad early choices can't be compensated later. Arguably, it can be due to interactions with the learning rate scheduler used and a more appropriate schedule could help solve this issue (although it is not clear what such a schedule should be).

### 4.2 CHECKPOINTS VS HYPERPARAMETERS - SELECTION BIASES

For PBT optimizers, one critical question is how much of the loss difference between two individuals is caused by different hyperparameters, and how much about different checkpoints. Both contributions are tightly entwined making it harder to identify which hyperparameters are best.

A naive initial option to counter this is to always start a PBT step from the best checkpoint in the population. In our experiments this performed worse, at least because of the noise in the evaluation metric, but also because of the short-term long-term discrepancy detailed above (Section 4.1). We also expect that doing so could make optimizers less robust by getting trapped in local minima too easily, or aggressively discarding more promising models in the longer term.

On the opposite side of the spectrum, we experimented with never culling checkpoints, effectively performing $n$ **full** trainings in parallel. Conceptually, keeping checkpoints is attractive since it should

add robustness to the optimizer: selected hyperparameters have to work well for more than one particular checkpoint. Indeed, this way the performance can be attributed to actual parameter schedules fitting different trainings, instead of being biased by checkpoints culling/random restarts. It also adds more variability which could be beneficial especially with respect to the short-term long-term discrepancy. That being said, we have neither observed significant improvement nor deterioration when keeping checkpoints, as long as the mutation schemes were not biased towards the best set of hyperparameters (e.g.: removing the $x^{\text{best}}$ term in Eq. 1), because doing so can make all hyperparameters converge towards the best checkpoint, making the optimization process early converge to values which are not necessarily adapted to other checkpoints.

Still, even with ROMUL's loss-agnostic mutation scheme, some checkpoints were observed to fall behind and waste resources if not culled, so we expect that a trade-off like the one we implemented (killing checkpoints after 3 failed mutations in a row) is necessary.

Another source of selection bias is noise. While the trainings are well behaved in PTB because the shuffling of the training set is synchronized by epoch, the trainings in CIFAR are much noisier because of the randomness introduced by data augmentation and the very small training set size. PBT optimizers based on more noise-robust blackbox optimization methods could be beneficial, but it is not clear how to adapt them.

## 5 RELATED WORK

Several families of methods exist for tuning hyperparameters of neural networks. Methods closest to grid search like random search (Bergstra & Bengio, 2012) and ASHA (Li et al., 2018) are based on minimal constraints and can be parallelized extensively. Methods striving for more data-efficient search (Bergstra et al., 2011; 2013; Feurer & Hutter, 2019) are more sequential in nature, requiring convergence of some trainings before launching new ones. Population-based training approaches (Jaderberg et al., 2017; Ho et al., 2019; Li et al., 2019) loosen the requirements of training different models, as hyperparameters are changed on-the-fly during training, which also makes the search for schedules easier and less structured, i.e. not based on a predefined function.

Recent advances in automatic discovery of data augmentation policies include Population Based Augmentation (Ho et al., 2019) which we compared to in this paper (denoted Truncation Selection). Another line of work on structuring the hyperparameter space for data augmentation policy search is AutoAugment (Cubuk et al., 2019a), FastAutoAugment Lim et al. (2019) and RandAugment Cubuk et al. (2019b), the later being faster and reaching top performance on CIFAR.

PBT is used successfully in reinforcement learning (Jaderberg et al., 2017), providing diversity in self-play and progressive difficulty, so other experimental comparisons that we did include Initiator PBT from (Li et al., 2019), which presented a generic PBT setup that inspired `hoptim`. For non-PBT baselines we used random search (Bergstra & Bengio, 2012), and ASHA (Li et al., 2018), which is an update on HyperBand (Li et al., 2017).

## 6 CONCLUSION

We introduced ROMUL, a robust PBT algorithm that we benchmarked on standard datasets with multiple regularization and data augmentation hyperparameters. Its main strength comes from its robustness to hyperparameters definitions by automatically adapting to the scale of each parameter. Although it did not show better performance on CIFAR than PBA – that was tuned for this benchmark – we demonstrated that it is more robust to domain changes. More importantly for the practical use-cases, it constitutes a good default that does not require extensive tuning to work well. We open-sourced its implementation as well as a simple and broadly compatible PBT library.

The main difficulties we observed for PBT-based optimizers came from short-term vs. long-term effects: parameters can have a positive impact in the short term but a negative one in the longer term which may not be rectifiable. Learning rate falls in this category, since decreasing it often provides quick gains at the risk of being trapped in a local minimum. Studying how to deal with such behaviors is in our opinion the main challenge of future work.

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

# A APPENDIX

## A.1 ROSENBROCK EXPERIMENTS

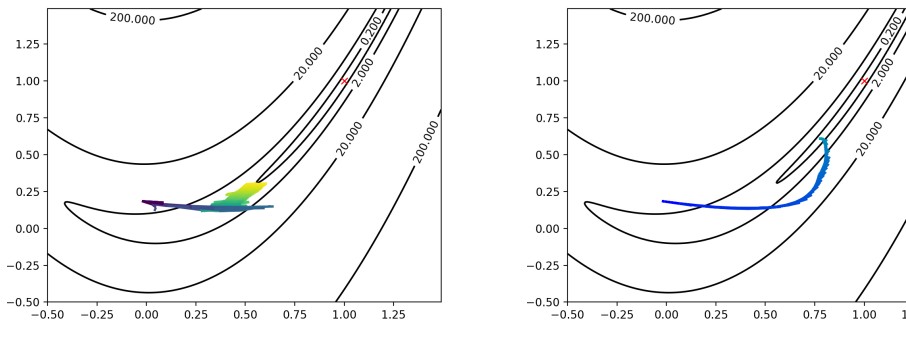

(a) Initiator PBT (minimum loss=0.20)          (b) Initiator PBT, small steps (minimum loss=0.048)

Figure 2: Trajectories of 100 Initiator PBT training steps (16 jobs per step) on the Rosenbrock function with $a = 1$ and $b = 100$ (minimum at the red cross $(1, 1)$, trajectories go from blue to green)

| Training | mean (log10) | std (log10) |
|---|---|---|
| Initiator (0.8/1.2 mult. steps) | -1.18 | 0.045 |
| Initiator (big steps) | -0.707 | 0.069 |
| Initiator (small steps) | -0.992 | 0.143 |
| ROMUL | **-2.101** | 0.678 |
| Truncation Selection | -0.834 | 0.327 |

Table 3: Final loss (in log10) mean and standard deviation for independent runs on the Rosenbrock testbed, computed over 20 runs (two-sample Welsh's test provides $p < 1.1e - 5$ when comparing each algorithm with ROMUL).

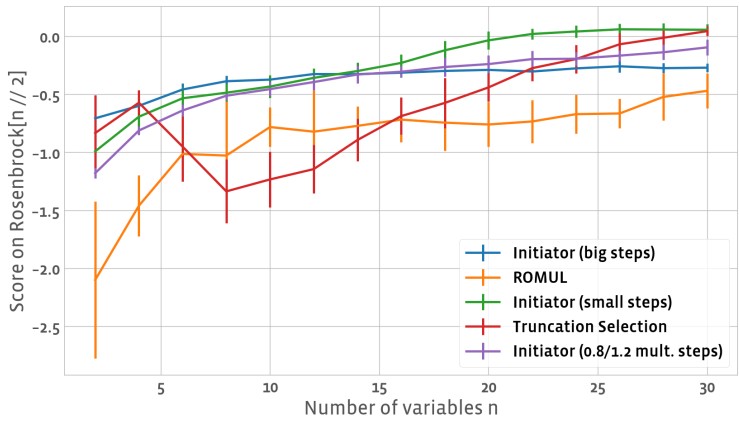

Figure 3: Score on the multi-variate Rosenbrock benchmark (explained in 3.1) over 20 experiments for each point. Lower is better, standard deviations are indicated. ROMUL performs well across the board for a wide range of number of variables, being surpassed only by Truncated selection in some regime ($n \in [\![8 \ldots 14]\!]$).

Table 4: The dropouts from Transformer-XL that we tune through PBT.

| | |
|---|---|
| dropouta | applied to multi-head attention layers |
| dropoute | to remove words from embedding layer |
| dropoutf | applied to positionwise ff layers |
| dropouti | for input embedding vectors |
| dropouto | applied to the output (before the logit) |

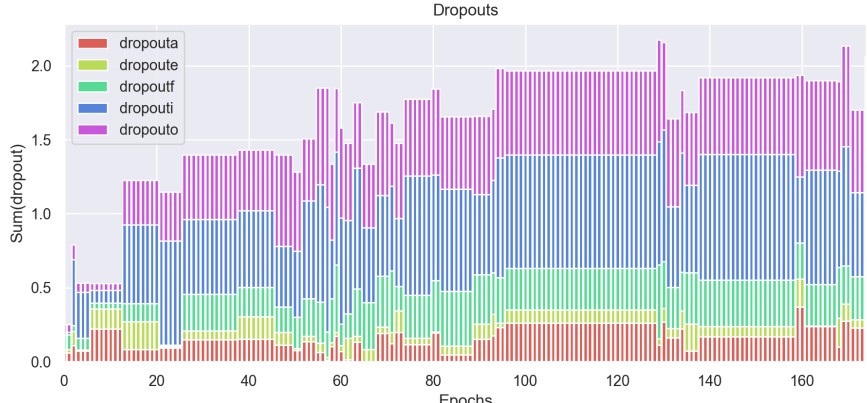

Figure 4: Dropout schedule of the best run of ROMUL 32 workers on PTB

## A.2 Language modeling on Penn Tree Bank

## A.3 Regularization schedules on Wikitext-103

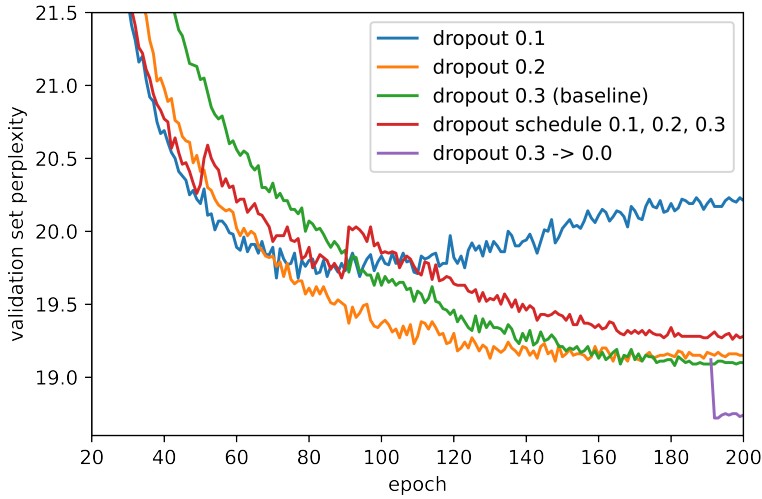

Figure 5: Lower dropout values are better early, but are outperformed by more strongly regularized models later (red, orange and blue lines) - here on wikitext103 with a 247M parameters language model from Fan et al. (2019) (*Adaptive Inputs + LayerDrop*). PBT algorithms would tend to reduce dropout aggressively early on: after that, even if the dropout is increased later, the performance remains worse than training with a high dropout from the beginning (red line). Perhaps counterintuitively, this hints against increasing regularization over the course of the training - in the opposite, we observe that fine-tuning the model without dropout significantly improves test performance (purple line reaches 17.98 test perplexity) compared to the baseline (green: 18.42 test perplexity)

A.4   SLIDES FOR INTERNAL PRESENTATION

| Training | Parallelism | Validation PPL | Test PPL |
|---|---|---|---|
| TransformerXL SOTA | 1 | 59.65 | 55.43 |
| ASHA | 16 | 63.20 | 58.35 |
| Truncation Selection PBT | 16 | 60.24 | 57.29 |
| Initiator PBT | 16 | 59.42 | 55.80 |
| ROMUL PBT | 16 | **57.83** | **55.16** |

Table 5: Perplexity (lower is better) on PTB for a Transformer-XL with 16 layers and 24M parameters

| Algorithm | CIFAR-10 | CIFAR-100 |
|---|---|---|
| Baseline: Wide-ResNet-28-10 | 3.9 | 18.8 |
| RandAugment (Cubuk et al., 2019b) | 2.7 | 16.7 |
| PBA (3 epochs/step) (Ho et al., 2019) | 2.6 | 16.7 |
| ASHA | 2.8 | 17.6 |
| Truncation Selection (PBA, ours) | 2.7 | 17.7 |
| Initiator PBT (Li et al. (2019), ours) | 2.9 | 17.9 |
| ROMUL | 2.8 | 17.1 |

Table 6: Classification error (lower is better) on CIFAR-10 and CIFAR-100 test sets for a Wide-ResNet-28-10 (36M params)

