# OpenReview forum: "ROMUL: Scale Adaptative Population Based Training"
_ICLR.cc/2021/Conference — Reject_

### Official Review · AnonReviewer1 · 2020-10-15
**Interesting problem, unconvincing solution.**

**Rating:** 4
**Confidence:** 4

**Review:**

** Summary **

This paper focuses on issues in the popular PBT algorithm for hyperparameter optimization. It investigates the 1) step size (which is typically a constant multiplier) 2) the variance induced by better weights and 3) the greediness of the algorithm, which they refer to as short-term vs. long term effects. These issues are well motivated, and it is intuitive that they are flaws in the original algorithm. The proposed approach is to use Differential Evolution which the authors claim makes the hyperparameter selection more robust. The paper also introduces a new library for online hyperparameter tuning.

** Primary Reason for Score **

The strengths of this work are that it identifies and discusses some interesting issues with PBT, a commonly used algorithm. However, as someone who frequently uses variants of the PBT algorithm, the evidence provided in this work is not sufficient for me to adopt their recommendations. The method is based on heuristics and the experiments are unfortunately not rigorous: the gains are small and it is a single seed. To increase my score, I would need to see more robust results that make these heuristics convincing, for example multiple seeds with clear outperformance (ideally statistically significant). It would also be important to see ablation studies for the newly introduced parameters (e.g. m). In addition, some demonstration of the phenomena described having an influence on the performance would be helpful.

** Strengths **

1) The issues the paper addresses are well motivated, and well described.
2) The topic of the paper (PBT) is one that I think has not been sufficiently addressed by the community. In particular, the present PBT algorithm is commonly used but none of the improvements since 2017 have been widely adopted. It seems like a fruitful direction for research.
3) I appreciate the discussion of the results, which do not claim SoTA but instead go into detail on possible drivers of performance.

** Weaknesses **

1) The main contribution of the work is not convincing. It simply replaces one heuristic for another. While the results show an improvement, it is not clear.
2) Experiments are only run a single time, and this is surely a noisy process. Given that, the gains vs. PBT seem small. It is entirely possible that this small gain is reversed in a second run. If the TransformerXL is too expensive, then a smaller experiment which can be repeated multiple times would be a stronger piece of evidence for the method’s efficacy.
3) The authors claim to reduce the meta-parameters, yet introduce new parameters (F_1, F_2 and m). Also how was the size of the PBT step chosen? For the transformer experiment it goes from 1 epoch -> 10 epochs. Some ablation studies for these parameters would be needed for a reader to fully understand how to use this method on a new task.
4) The library is presented as a second major contribution, but it is not clear why the reader would choose to use it over existing libraries such as ray tune, which are popular and widely used. There is no comparison or discussion here, other than just saying that the new library is better. I also couldn’t find the library anywhere, the supplementary material is just a two page pdf, and there is no anonymized link. Please correct me if I missed this.

** Minor issues **

i) The ICLR 2020 template was used (rather than 2021).

ii) Bottom of page 7, “raw” -> “row”

---

> ### Author Response · Authors · 2020-11-20
> **review answer (1/2)**
>
> Thank you for your review
> > The method is based on heuristics and the experiments are unfortunately not rigorous: the gains are small and it is a single seed. To increase my score, I would need to see more robust results that make these heuristics convincing, for example multiple seeds with clear outperformance (ideally statistically significant).
>
> > Experiments are only run a single time, and this is surely a noisy process. Given that, the gains vs. PBT seem small. It is entirely possible that this small gain is reversed in a second run. If the TransformerXL is too expensive, then a smaller experiment which can be repeated multiple times would be a stronger piece of evidence for the method’s efficacy.
>
> We understand the concern about significance of the results, and it’s an important point when evaluating methods of hyperparameter tuning. In practice, our experiments have dozens of workers in parallel, and it’s been challenging to provide sensitivity analysis on the neural network trainings given the significant resource requirements (24 GPUs x 12 hours for one experiment on average). Note however that on PTB we provide 2 runs (one with 16 workers and the other with 32) with consistent behaviors, even though it is definitely not enough for statistically significant evidence.
> To partially overcome this lack of statistical evidence, we repeated the Rosenbrock benchmark on 20 runs, and performed a t-test on the results to assess the signicativity of the difference. The p-value was below 1e-4 highlighting that ROMUL performs significantly better than other methods on this ill-conditioned problem. These results are now included in section 3.2 of the paper and in the Table 3 of the appendix.
>
> > The main contribution of the work is not convincing. It simply replaces one heuristic for another. While the results show an improvement, it is not clear. It would also be important to see ablation studies for the newly introduced parameters (e.g. m). In addition, some demonstration of the phenomena described having an influence on the performance would be helpful. The authors claim to reduce the meta-parameters, yet introduce new parameters (F_1, F_2 and m).
>
> Neither F1, nor F2 nor m are modified throughout all of the experiments, so they should not be considered as parameters but as constants which a user does not have to care about. Users only need to provide bounds for the parameters, and the scaling of the updates automatically adapts during the course of the training. This is a major difference compared to other optimizers, for which mutation step sizes need to be carefully provided. Indeed, this mutation step size has a major impact on the optimization as can be seen through the Rosenbrock benchmark: different step sizes in Initiator PBT lead to an order of magnitude improvement (see the newly added table 3 in the Appendix). When such scaling is unknown, or if there are many hyperparameters, tuning it manually is in our opinion the major difficulty of current PBT algorithms since good parametrization on one application cannot be straightforwardly transferred to another application. While ROMUL does not always reach the performance of hand-tuned PBT algorithms for specific applications, we aimed at showing it was good on a broad range of settings, without manual tuning.
>
> > Also how was the size of the PBT step chosen? For the transformer experiment it goes from 1 epoch -> 10 epochs. Some ablation studies for these parameters would be needed for a reader to fully understand how to use this method on a new task.
>
> We have indeed limited our experiments on trainings with around 100 to 300 epochs, and 1 step per epoch to obtain somehow similar trainings. This is now explicitly mentioned in the beginning of the experiment section (Section 3). The experiment with 10 epochs on PTB was only to highlight and try to understand the difficulties which arose during the training.

---

> > ### Author Response · Authors · 2020-11-20
> > **review answer (2/2)**
> >
> >
> > > The library is presented as a second major contribution, but it is not clear why the reader would choose to use it over existing libraries such as ray tune, which are popular and widely used. There is no comparison or discussion here, other than just saying that the new library is better. I also couldn’t find the library anywhere, the supplementary material is just a two page pdf, and there is no anonymized link. Please correct me if I missed this.
> >
> >
> > The package is indeed not yet open sourced but will be when we make the article public.
> > Compared to Ray Tune:
> >  - Hoptim is simpler, standalone for PBT, independent of any scheduler, while Ray Tune depends on Ray (API for distributed training, with a central server).
> >  - Experiments can be resumed and agents can be added to the population.
> >  - Hoptim has multiple optimization benchmarks and training use cases that come with the library.
> >
> > Overall, we have had issues with ray on a slurm cluster and wanted to keep full flexibility with hoptim both in term of research (being able to master every piece of it) and of scheduling (split the workers through several jobs, adding/removing workers, handling preemptions). The current design is completely decoupled from the scheduling part, it does not assume a slurm cluster but does require a shared file system. Eventually, we expect that ROMUL can be easily ported to Ray Tune anyway.
> >
> > > The ICLR 2020 template was used (rather than 2021).
> > > Bottom of page 7, “raw” -> “row”
> >
> > Thank you for noticing, this is updated.

---

> > > ### Comment · AnonReviewer1 · 2020-11-22
> > > **Suggestions to improve the paper...**
> > >
> > > Hi - I am just writing to say that I have read the comments and will not be changing my score. In saying that, there are elements of the work which seem promising so below are my two main suggestions to improve it for future submission (I suspect it will take >> 1 week).
> > >
> > > The experiments are not at all convincing. This does not mean they need to be *larger*. In fact, it would be great if there was an intermediate experiment, where PBT does well, such as a small RL task. The Rosenbrock is good to motivate/explain but it is not enough to convince, and so you need something slightly larger which can produce robust and intuitive results. Then after that, you can include the larger parts, but I don't trust results with one seed so alone would have to discount those.
> > >
> > > I am also not convinced that the library should be introduced as a contribution, unless there is a strong empirical reason why it is better than ray tune, who clearly have a well-functioning PBT algorithm (I don't work for Ray). I think your paper would be better if you focus more on the findings regarding why PBT fails and how your proposed solution solves this problem, rather than confounding it by discussing a library which is very hard to evaluate. The presence of the open source code is of course welcome, but as a supplement.

---

### Official Review · AnonReviewer3 · 2020-10-26

**Rating:** 4
**Confidence:** 4

**Review:**

This submission studies Population Based Training (PBT) methods for tuning and adapting hyperparameters over the course of training. It makes two contributions: (1) a novel PBT algorithm, ROMUL, and (2) a library for PBT-based training, hoptim.

After reading the paper and the supplementary material, I can only comment on the first contribution. While the authors highlight the hoptim library as one of the main contributions of the paper, they do not describe what are their advantages over existing hyperparameter optimization frameworks or existing PBT implementations (e.g. the one in Ray). As far as I can tell, there is no source code or link to an anonymous repository so that we can evaluate this contribution. Commands for replicating experiments using hoptim are scattered throughout the manuscript, but please note that this is not enough to evaluate the quality or impact of the software. This is an important issue because the paper justifies weaker empirical results based on implementation differences that are never discussed (e.g. “The differences in the job and population management in hoptim may explain the difference between our implementation and theirs, which is particularly marked on the training set reduced CIFAR-10: 12.8% for their vs. 13.9% for our implementation.”, or how the number of workers has a strong impact in the final result for PTB experiments.).

By leveraging ideas from Differential Evolution, ROMUL eases the task of defining the search space when considering hyperparameters with different magnitudes. In other words, this simplifies the task of “tuning the hyperparameter tuner”. The benefits of the proposed strategy are showcased by optimizing a 2D Rosenbrock function where the optimal values for the two parameters differ in magnitude (a=1, b=100). ROMUL is then applied to optimize Population Based Augmentation (PBA) on a reduced training set for CIFAR-10 and to tune the dropout rates in TransformerXL for Penn Treebank (PTB). Results for baseline methods are slightly worse than those in the literature even when using the code provided by the authors, and authors are encouraged to explain the reason for this. ROMUL seems to outperform other PBT methods when tuning TransformerXL, but the benefits on PBA when applied to CIFAR-10 are not so clear. It is difficult to evaluate the significance of these figures, as no standard deviation across seeds is reported.

My main concern regarding the experimental setup has to do with the mutation constant used for Initiator PBT and Truncation PBT. The works by Li et al. (2019) and Jaderberg et al. (2017) perturb hyperparameters by a multiplicative factor of 1.2 or 0.8 instead. This enables a much finer-grained search space than the one implemented in this submission, where the additive mutation constant might be too large for some parameters given the range in which these parameters are defined. For instance, the optimal value of $a$ in the Rosenbrock function is 1 but the step size for Initiator PBT is $(hi-lo)/30=224.24/30=7.47$. Since $\hat{a}$ is initialized to 20, it is impossible for this method to even get close to the optimal value.

The discussion section provides some interesting experiments showcasing the effect of some design choices on PBT methods. It discusses the importance of the patience of the algorithm in order to account for the long-term impact of some hyperparameters (e.g. learning rate, dropout rate) as well as the impact of reusing existing checkpoints after mutation.

While this submission discusses important research topics, I do not believe it is ready for publication yet. The authors highlighted two main contributions, but I believe there are three potential ones: (1) a PBT method that does not need extensive tuning, (2) a software library for PBT training, and (3) an empirical evaluation of different design choices for PBT methods. However, these need to be developed further (and potentially in separate papers) before they can be published at ICLR.

---

> ### Author Response · Authors · 2020-11-20
> **Thank you for your review**
>
> Thank you for your review
>
> > While the authors highlight the hoptim library as one of the main contributions of the paper, they do not describe what are their advantages over existing hyperparameter optimization frameworks or existing PBT implementations (e.g. the one in Ray). As far as I can tell, there is no source code or link to an anonymous repository so that we can evaluate this contribution.
>
> The package is indeed not yet open sourced but will be when we make the article public.
> Compared to Ray Tune:
>  - Hoptim is simpler, standalone for PBT, independent of any scheduler, while Ray Tune depends on Ray (API for distributed training, with a central server).
>  - Experiments can be resumed and agents can be added to the population.
>  - Hoptim has multiple optimization benchmarks and training use cases that come with the library.
>
> Overall, we have had issues with ray on a slurm cluster and wanted to keep full flexibility with hoptim both in term of research (being able to master every piece of it) and of scheduling (split the workers through several jobs, adding/removing workers, handling preemptions). The current design is completely decoupled from the scheduling part, it does not assume a slurm cluster but does require a shared file system. Eventually, we expect that ROMUL can be easily ported to Ray Tune anyway.
>
> > Results for baseline methods are slightly worse than those in the literature even when using the code provided by the authors, and authors are encouraged to explain the reason for this.
>
> The official github repo (https://github.com/kimiyoung/transformer-xl) does not contain the script for training with Pytorch on PTB. We used the script mentioned in this thread https://twitter.com/ZihangDai/status/1245905407350112256 by the first author of Transformer-XL, which is in github: zihangdai.github.io/misc/ptb.zip. The example_log.txt file in this zip states provides “test ppl 55.60” at the end of the training, which we do replicate (we obtain 55.43).
> > ROMUL seems to outperform other PBT methods when tuning TransformerXL, but the benefits on PBA when applied to CIFAR-10 are not so clear. It is difficult to evaluate the significance of these figures, as no standard deviation across seeds is reported.
>
> We understand the concern about significance of the results, and it’s an important point when evaluating methods of hyperparameter tuning. In practice, our experiments have dozens of workers in parallel, and it’s been challenging to provide sensitivity analysis on the neural network trainings given the significant resource requirements (24 GPUs x 12 hours for one experiment on average). However, we repeated the Rosenbrock benchmark on 20 runs, and performed a two sample Welch t-test on the results to assess the signicativity of the difference. The p-value was 1.4e-5, meaning that ROMUL performs statistically significantly better than other methods on this ill-conditioned problem. These results are now included in section 3.2 of the paper and in the Table 3 of the appendix.
>
> > My main concern regarding the experimental setup has to do with the mutation constant used for Initiator PBT and Truncation PBT. The works by Li et al. (2019) and Jaderberg et al. (2017) perturb hyperparameters by a multiplicative factor of 1.2 or 0.8 instead. This enables a much finer-grained search space than the one implemented in this submission, where the additive mutation constant might be too large for some parameters given the range in which these parameters are defined.
>
> This is true that such multiplicative factor allow reaching values arbitrarily close to the optimal ones, however, this is still lacking because:
> while the steps can “reach” closer values, it does not mean they “converge” to this values, since they will still have “close to” fixed steps around the optimal value.
> Such multiplicative steps assume a logarithmic dynamic, which may be inaccurate, hence large values would always have larger step than small values. For dropouts for instance, this does not seem accurate since we would probably need smaller steps on both low and high dropouts.
> These values may still need to be adapted for each specific application, which is impractical at scale.
> Overall, this comes back to the adaptability requirement that has led to ROMUL: without step adaptation mechanism, it is not possible to converge to optimal solutions.
> In any case, we have rerun the Rosenbrock benchmark with multiplicative steps for Initiator PBT, reaching results in between the big steps and the small steps, leaving still an order of magnitude gap compared with ROMUL. This has been added in Table 3 in the Appendix.

---

> > ### Comment · AnonReviewer3 · 2020-11-22
> > **Reply**
> >
> > Thanks for your reply and for clarifying some of the points that were unclear in the paper.
> >
> > I appreciate the effort in running more seeds for some of the experiments and providing results for PBT with multiplicative steps. However, I agree with AnonReviewer1 in the fact that while results on the Rosenbrock benchmark might make a good motivating example, it is not enough to prove that the proposed method outperforms existing population-based approaches when it comes to optimizing neural networks. The paper would benefit from statistically significant results on problems involving neural networks, which do not need to be as large scale as the ones in this submission. In that case, it would be fine to provide single seed results for the large scale experiments reported in this submission.
> >
> > I believe that running such experiments is out of the scope of the rebuttal phase and will require major changes to the paper. For this reason, I will keep my original rating. As I wrote in my review, I believe this paper studies some important research directions and I would suggest to include the proposed changes in a future submission.

---

### Official Review · AnonReviewer2 · 2020-10-28
**Motivation of the proposed method is not clear and the writing can be further improved**

**Rating:** 3
**Confidence:** 4

**Review:**

In this submission, the authors propose a modification to the PBT (population-based training) method for HPO. It is interesting, however, there are several important issues to consider:

1) The major part of the proposed method ROMUL is to replace some update rules based on Differential Evolution, which is a well-studied method in Evolutionary Algorithms. The novelty of the proposed method ROMUL is not high. But more important, it is unclear why such modifications are necessary. In other words, what new challenges in HPO can be addressed by conducting these modifications to PBT. Without clear and strong reasons to motivate these modifications, it is hard to evaluate the proposed method.

2) The writing of this submission can be further improved. Many paragraphs and sentences are not logically organized, and it is difficult to understand the main points of the submission. For example, based on the Introduction section, it seems that the main part of this submission is to "empirically study the different training dynamics of ..." (second paragraph). And in introduction, the authors didn't well motivate the proposal of their method. Although several challenges are mentioned in the first paragraph, it is not clear which ones are solved by the proposed method, and how they are tackled.

Personally I feel that "fixed step" issue in PBT is important, which should be mentioned early.

Several interesting findings are provided in the experiments. The authors can make them clear and highlight them by improving the writing of current version.

---

> ### Author Response · Authors · 2020-11-20
> **Thank you for your review**
>
> Thank you for your review
>
> > The major part of the proposed method ROMUL is to replace some update rules based on Differential Evolution, which is a well-studied method in Evolutionary Algorithms. The novelty of the proposed method ROMUL is not high. But more important, it is unclear why such modifications are necessary. In other words, what new challenges in HPO can be addressed by conducting these modifications to PBT. Without clear and strong reasons to motivate these modifications, it is hard to evaluate the proposed method.
>
> The Differential Evolution (DE) approach is indeed a well studied method for black box optimization, which works pretty well in the context of finding optimal hyperparameters. It’s based on a population of individuals, but is not a “Population Based Training” (PBT) algorithm, as we define it in the context of our paper, which solves a different problem.
> In (standard) DE, we train N models to convergence, report their loss, train N other models suggested by DE, and repeat the process T times. Because several iterations are required, DE does not converge before `T` times the wall clock time of a single training at best. Romul does not follow this scheme.
> In PBT approaches (including our approach Romul adapted from DE), we evaluate the models after they have done N/T of their training. Thus, we aim at finding a good model with good hyperparameters within the wall clock time used to train a single model. Our approach also finds schedules over hyperparameters, rather than fixed values, which are shown to improve models performance.
> Fitting a restricted compute budget, and the need for schedules over some hyperparameters (eg data augmentation) are key reasons why practitioners use PBT approaches (e.g. Romul) rather than regular hyperparameter optimization methods (e.g. DE).
>
> Overall we consider it is still a variant of DE, but explain the adaptations required to make it work in the context of population based training. As shown in section 2.2, the differences include adding the concept of checkpoint, since this is a specificity of population based training compared to standard derivative-free optimization problems, and handling a dynamic function which strongly biases pairwise comparisons with parents in DE.
>
> > Personally I feel that "fixed step" issue in PBT is important, which should be mentioned early.
>
> This is indeed one of the most important feature of ROMUL compared to current PBT methods. While this was mentioned in the abstract, it was indeed not explicitly stated in the introduction. We added a couple of mention of it.

---

### Official Review · AnonReviewer4 · 2020-10-30
**Good paper**

**Rating:** 6
**Confidence:** 5

**Review:**

#### Summary
The paper provides a new variant of PBT which utilizes ideas from differential evolution and cross-over. The original PBT and even initiator PBT do not perform crossover on the hyper-parameters, and insufficient cross-over may cause PBT to perform greedy in the initial phases which ends up with a suboptimal convergence. The investigation of better cross-over in PBT is itself an interesting research direction and the authors demonstrated its effectiveness in standard benchmarks and data augmentation tasks. The improvements of ROMUL-PBT are also helpful to the community since PBT has been applied in a variety of real world applications.

#### Pros
1. Quality: The paper quality is in general good. The experiments are well designed and the results are good. So the experiments clearly supports the argument that differential evolution helps PBT.
2. Clarity: The paper is well written and easy to follow. The organization is also clear.
3. Originality: I think that adapting ideas from differential evolution to PBT is new, even though differential evolution itself is not something new.
4. The paper provides some benchmarking of PBT related algorithms in image classification, language modeling and data augmentation which is good for the community to understand these approaches.


#### Cons
1. Significance: the improvements over existing methods seem slight. The experiments do not provide sensitivity analysis so it is a bit hard to conclude whether the results are statistically significant. But at the same time, the proposed method does show promise.
2. As a thorough evaluation purpose, it would be interesting to see how the proposed methods work in large set of hyperparameters (magnitude of 10-100).

#### Questions
1. PBT needs to use validation loss to obtain fitness. Is your result evaluated on the validation data or the test data? If only evaluating on the validation data, the result may not reveal potential overfitting to the validation set. So it would be nice to have results on a held-out test set.

---

> ### Author Response · Authors · 2020-11-20
> **Thank you for your review**
>
> Thank you for your review.
>
> > Significance: the improvements over existing methods seem slight. The experiments do not provide sensitivity analysis so it is a bit hard to conclude whether the results are statistically significant. But at the same time, the proposed method does show promise.
>
> We understand the concern about significance of the results, and it’s an important point when evaluating methods of hyperparameter tuning. In practice, our experiments have dozens of workers in parallel, and it’s been challenging to provide sensitivity analysis on the neural network trainings given the significant resource requirements (24 GPUs x 12 hours for one experiment on average). However, we repeated the Rosenbrock benchmark on 20 runs, and performed a two sample Welch t-test on the results to assess the signicativity of the difference. The p-value was below 1e-4 highlighting that ROMUL performs significantly better than other methods on this ill-conditioned problem. These results are now included in section 3.2 of the paper and in the Table 3 of the appendix.
>
> > As a thorough evaluation purpose, it would be interesting to see how the proposed methods work in large set of hyperparameters (magnitude of 10-100).
>
> Following the same reasoning as above, we can experiment on multiple Rosenbrock, with the drawback that, again, this may not be representative of actual trainings. Interestingly, ROMUL still performs best or second best over other optimizers up to around 30 parameters. This will need to be investigated further in future works.
>
> > PBT needs to use validation loss to obtain fitness. Is your result evaluated on the validation data or the test data? If only evaluating on the validation data, the result may not reveal potential overfitting to the validation set. So it would be nice to have results on a held-out test set.
>
> We want to clarify this: we train on the training set, and validate on the validation set. This validation loss is reported to the PBT algorithm. At the end of the training, we evaluate the models on the test set, and report the results on our paper (See Tab. 2 containing only test set results, and Tab. 3 providing both validation and test set results).
> The overfitting issue  is however still an interesting point, and indeed in our early experiments we found out that PBT algorithms could all significatively overfit the validation set (some more than others, mainly depending on how “greedy” they are in selecting good checkpoints although we did not back this observation with experiments, cf Section 4.2).

---

### Author Response · Authors · 2020-11-20
**Thank you for productive feedback**

Thanks to the reviewers for productive feedback. We added some experiments on Rosenbrock to show the behavior of the various PBT algorithms:
1. in statistical robustness across several runs,
2. For more hyperparameters (larger number of variables optimized by PBT).

---

### Decision · Program_Chairs · 2021-01-07
**Final Decision**

**Decision:**

Reject

**Comment:**

This submission proposes a variant of population based training (PBT) for hyperparameter selection/evolution, aimed at addressing drawbacks of existing variants (e.g. the coupling of the choice of checkpoint with the choice of hyperparameters). Reviewers generally agreed that the paper is interesting and covers an important topic, and the evaluation does show improvements over existing PBT variants. On the other hand they also raised a few important issues:

1. The `hoptim` library is claimed as a primary contribution of the work, but it is not clear from the manuscript what benefits this library offers over existing software. When claiming a library as a main contribution, it is helpful to provide a more thorough description of the software and its benefits, and/or ideally a link (anonymized for review) to the software. The authors did respond by providing a brief description of the benefits of the library, mitigating this issue somewhat. However it's still difficult to discern how/whether to weigh the open source library as a main contribution of the paper.

2. The evaluation is not very convincing: the differences are small and error margins are not provided for the neural network-based experiments, meaning that any differences could be due to noise. The authors fairly point out that it is difficult to perform multiple runs of these experiments as the resource requirements are large, and they have done 20 runs of the Rosenbrock experiment with smaller compute requirements. But the reviewers were not convinced that the Rosenbrock experiment reflects the method's application to neural network hyperparameter selection; the problems are too different. The submission would be significantly stronger if it included results over multiple runs of an "intermediate" sized experiment on a problem involving a neural network demonstrating that ROMUL outperforms competing approaches by a statistically significant margin.

3. The proposed approach is ultimately heuristic. This is not necessarily a problem if there are strong empirical results demonstrating the efficacy of the proposed heuristic, but in this case the empirical results didn't convince (see point 2).

Given these concerns raised by reviewers, the submission is not quite ready for ICLR. I hope the authors will consider resubmitting the paper after improving it based on the reviewers' feedback.